# Parkin-mediated mitophagy is negatively regulated by FOXO3A, which inhibits Plk3-mediated mitochondrial ROS generation in STZ diabetic stress-treated pancreatic β cells

Ji Yeon Shim[1]°, Jin Ook Chung[2]°, Dawa Jung[3]°, Pil Soo Kang[3], Seon-Young Park[2], Ayse Tuba Kendi[4], Val J. Lowe[4]*, SeungBaek Lee [4,5]*

**1** College of Nursing, Dankook University, Cheonan, Chungcheongnam, Republic of Korea, **2** Department of Internal Medicine, Chonnam National University Medical School, Gwangju, Republic of Korea, **3** U&Hang Clinic, Asan, Chungcheongnam, Republic of Korea, **4** Division of Radiology, Mayo Clinic, Rochester, Minnesota, United States of America, **5** Department of Molecular Pharmacology and Experimental Therapeutics, Mayo Clinic, Rochester, Minnesota, United States of America

☯ These authors contributed equally to this work.
* vlowe@mayo.edu (VJL); lee.seungbaek@mayo.edu (SBL)

**Data Availability Statement:** All relevant data are within the paper and its Supporting Information files.

## Abstract

Diabetes mellitus (DM) is one of the most researched metabolic diseases worldwide. It leads to extensive complications such as cardiovascular disease, nephropathy, retinopathy, and peripheral and central nervous system through an inability to produce or respond to insulin. Although oxidative stress-mediated mitophagy has been reported to play an important role in the pathogenesis of DM, specific studies are still lacking as well as remain highly controversial. Here, we found that Parkin-mediated mitophagy in pancreatic β cells under streptozotocin (STZ)-diabetic stress was induced by Polo-like kinase 3 (Plk3) and inhibited by the transcription factor Forkhead Box O3A (FOXO3A). STZ stress induces mitochondrial recruitment of Parkin through Plk3-mediated mitochondrial reactive oxygen species (ROS) generation, which causes pancreatic cell damage. Conversely, FOXO3A acts as negative feedback to prevent diabetic stress by inhibiting Plk3. Meanwhile, antioxidants including N-acetylcysteine (NAC) and natural COA water scientifically block these mitochondrial ROS and mitochondrial recruitment of Parkin by inhibiting Plk3. Through a 3D organoid *ex vivo* model, we confirmed that not only ROS inhibitors but also mitophagy inhibitory factors such as 3-MA or Parkin deletion can compensate for pancreatic cell growth and insulin secretion under STZ diabetic stress. These findings suggest that the Plk3-mtROS-PINK1-Parkin axis is a novel mitophagy process that inhibits pancreatic β-cell growth and insulin secretion and FOXO3A and antioxidants may provide new alternatives for effective diabetes treatment strategies in the future.

## 1. Introduction

Diabetes mellitus (DM) is a chronic metabolic disorder characterized by abnormally high blood glucose, which can lead to serious microvascular (diabetic kidney disease, diabetic

**Funding:** V.L., and S.B.L. supervised the project. This work was also supported by grant number E&P-RES20191001-02 to S.B.L. and by the Dorothea Berggren Charitable Foundation to V.L. The funders had no role in study design, data collection and analysis, decision to publish, or preparation of the manuscript.

**Competing interests:** Dr. Lowe is a consultant for AVID Radiopharmaceuticals, Eisai Co. Inc., Bayer Schering Pharma, GE Healthcare, and Merck Research, and receives research support from GE Healthcare, Siemens Molecular Imaging, AVID Radiopharmaceuticals, and NIH (NIA, NCI). Dr. Lee reported grants from E&P Co., Ltd. (South Korea) during the conduct of the study. This does not alter our adherence to PLOS ONE policies on sharing data and materials.

retinopathy, and diabetic neuropathy) and macrovascular (coronary heart disease, stroke, and peripheral artery disease) complications [1–18]. DM is a heterogeneous disease with a common feature of hyperglycemia [1, 3, 12, 19]. Pancreatic β-cell dysfunction resulting in a relative or absolute insulin deficiency is a key process in the development of DM [3, 6, 8, 9, 15, 18, 19]. In addition, residual β-cell function even in patients with preexisting DM might be associated with better clinical outcomes, including less hypoglycemia and microvascular complications [2–6, 8–15, 17–20]. Therefore, protecting the secretory function of β-cell is important from a clinical viewpoint for the treatment of DM [2, 3, 7, 11, 12, 18, 19]. Although many studies are being conducted on the development of therapeutic agents as well as the study of diabetes-related mechanisms, there are still very few alternatives to the causes and specific solutions. Streptozotocin (STZ) is one of the convenient agents for the study of diabetes-related mechanisms occurring in cells by direct treatment in animals and pancreatic cells [1, 9, 15–17].

Recently, oxidative stress is well known to be associated with the pathogenesis of metabolic diseases, including diabetes and complications [1, 8, 11, 12, 14, 19–21]. Increased ROS levels were observed in diabetic states, which are closely related to increased modified mitochondrial metabolites, altered mitochondrial proteins, and decreased expression of antioxidant enzymes [7, 10–12, 14, 19–21]. In the diabetic heart, mitochondria overproduce mitochondrial ROS by hyperglycemic conditions, which play an important role in the formation and development of diabetic cardiomyopathy (DCM) [4, 5, 10, 16, 22, 23]. Also, excessive amounts of glucose increase mitochondrial ROS in hyperglycemia-sensitive cells, which impairs mitochondrial function in the cells [11, 19, 24]. And ROS induces insulin resistance in the body by inactivating the mechanism of interaction between the insulin receptor and the glucose transport system [3, 24]. Diabetes is also a key regulator of atherosclerosis through ROS [10, 12]. Mitophagy, one of autophagy, is stimulated by decreased mitochondrial membrane potential (ΔΨm) and an increase in ROS [5, 21, 25–27]. This triggers the recruitment of PTEN Induced Kinase 1 (PINK1)-Parkin complex in the mitochondrial outer membrane [17, 21, 22, 25–29]. Upon mitochondrial depolarization, PINK1 accumulates in the outer membrane of mitochondria and induces phosphorylation of Ser65 of the E3 ubiquitin ligase Parkin, resulting in its translocation to the mitochondria [17, 21, 22, 27, 28]. Mitophagy plays a role in maintaining cellular metabolism by preventing oxidative stress in the diabetic heart [4, 5, 22, 23]. Expression of mitophagy, including autophagy, was markedly suppressed in the heart of the T1D animal model [6, 16, 30]. However there are recent many reports that the activation of mitophagy has a negative effect on DCM, high-fat diet (HFD) in patients with T2D, or hyposalivation of the submandibular gland in db/db mice, so there are still a lot of controversies [5, 17, 22, 23, 31]. Therefore, Parkin-mediated mitophagy research on diabetes requires various approaches and specific mechanism studies using *in vitro* cell studies, *in vivo* animal experiments, and *ex vivo* 3D organoid systems.

In this study, we found the mitochondrial translocation of Parkin in pancreatic β cells under STZ diabetes-induced stress. Parkin is affected by Plk3-induced mitochondrial ROS, and at this time, PINK1 phosphorylation results in the recruitment of Parkin into the mitochondria. Conversely, FOXO3A induced by antioxidants directly inhibits Plk3, thus suppressing the ultimate role of mitophagy in STZ-treated pancreatic cells. Although the mitophagy of Parkin has a negative effect on beta cell growth and insulin secretion, antioxidants and mitophagy inhibitors can inhibit Parkin translocation for mitophagy by inhibiting PINK1 activity.

## 2. Materials and methods

### 2.1. Cell lines and reagents

All cell lines were sourced from commercial vendors. Beta TC-6 and Beta TC-tet mouse islet cell lines (S1 Fig in S1 File) were cultured in Dulbecco's modified Eagle's media (DMEM) with

15% FBS and Streptomycin and Penicillin. The cells and reagents were prepared in the same way as in the previous study [9]. Briefly, this medium was sterilized immediately by filtration using a membrane with a porosity of 0.2 micrometers. Streptozotocin (STZ, CAS 18883-66-4) was purchased from Calbiochem. Carbonyl cyanide 3-chlorophenylhydrazone (CCCP, C2759), N-acetylcysteine (NAC, 616-91-1), Hydrogen peroxide ($H_2O_2$, 7722-84-1), and 3-methyladenine (3-MA, M9281) were purchased from Sigma-Aldrich. MitoSOX was obtained from Molecular Probes (Eugene, OR). Paraformaldehyde Solution (MFCD00133991) and 4',6-diamidino-2-phenylindole (DAPI, 62248) were purchased from Thermo Scientific.

## 2.2. Transmission electron microscopy

After STZ treatment, two Beta TC-6 and Beta TC-tet mouse pancreas islet cells were washed with 1xPBS at 4˚C, and then attached cells were separated from the culture dish with Trypsin-EDTA. The centrifuged pancreas β cells were fixed overnight in 2% glutaraldehyde in cacodylate buffer (0.1M sodium cacodylate, 2mM MgCl2) at 4˚C. Samples were washed at least 3 times with cacodylate buffer stored in a cold state, and then post-fixed in 2% osmium tetroxide for 1 hour. The samples were reacted with propylene oxide for solvent displacement and embedded in Eponate 812 (Agar Scientific, Stansted, UK) resin, and the samples were sectioned with an ultramicrotome. Sections were examined at 60 kV with a JEM-1200EX electron microscope (JEOL; Akishima, Tokyo, Japan) and there the organelles were observed and quantified.

## 2.3. Plasmids, gene silencing by siRNAs and lentiviral shRNAs

Parkin's constructs have been described previously [28, 29, 32]. GFP, Flag-tagged PINK1, or Flag-FOXO3A or were purchased from Addgene (Watertown, MA). Plk3 WT plasmid was generated by inserting the full-length Plk3 cDNA amplified from the pcDNA3-HA-Parkin plasmid. *FOXO3A* siRNAs were purchased from Invitrogen (Tokyo, Japan). Lentiviral-*Parkin* or *PINK1* shRNAs were manufactured by dividing the gene target area into three areas and detailed sequence information is described in section S1 Fig in S1 File.

## 2.4. Transient transfection, viral infection, and stable transduction

For transient overexpression studies, Beta TC-6 cells are first discarded of the existing serum-containing medium, rinsed 3 times with 1xPBS, and then replaced with a serum-free medium and left for 5 minutes. Then, the cells were transiently transfected with Flag or Flag-tagged Plk3 (or GFP-tagged PINK1 and Parkin) with a transfection reagent such as lipofectamine 2000 (Invitrogen). After 6 hours, discard the serum-free medium containing the transfection reagent and rinse again with 1xPBS. After changing the medium to which fresh serum has been added, the cells are cultured for 24 h, and then, gene expression is checked under a fluorescence microscope. For viral infection, Beta TC-6 cells were performed at a cell density of 60–70%, and the cell culture medium was pretreated with polybrene one day before. *PINK1*, *Parkin*, or *FOXO3A* lentiviral shRNAs (S1 Fig in S1 File) were infected with cells for 12 hours and then replaced with fresh medium. Cells infected with 2 μg/ml puromycin (Sigma-Aldrich) were selectively harvested. *Control* shRNA was simultaneously performed as a negative control. Infection or transfection of vectors was prepared in the same way as in the previous study [9, 28, 32–34].

## 2.5. Measurement of mitochondrial ROS production or Parkin mitochondria translocation using immunofluorescence assay

Beta TC-6 cells were treated with STZ (50 μM) for 12 h or Carbonyl cyanide 3-chlorophenyl-hydrazone (CCCP, 10 μM) for 4 h. Cells were washed 3 times with 1xPBS and then cells were fixed with 0.3% formaldehyde. And then cells were observed under a fluorescence microscope and quantified. For measurement of mitochondrial ROS production, cells were transfected with DNA vectors using lipofectamine for gene overexpression or infected with lentiviral vector for suppression of specific gene. Cells were transfected or infected with the indicated plasmids (Flag or Flag-PINK1, HA-Plk3, *Control* shRNA or *PINK1* shRNA) and then cells were treated with streptozotocin for 18 h or $H_2O_2$ (10 μM) for 12 h (sometimes, 10 mM NAC or natural COA water). Cells were stained with MitoSOX (5 μM) for 30 min and then cells were analyzed by immunofluorescence assay and quantified. For measurement of Parkin mitochondrial recruitment, GFP-Parkin overexpressed beta cells were transfected with or infected with the indicated plasmids (Flag or Flag-PINK1, HA-Plk3, *Control* shRNA or *PINK1* shRNA) and then cells were treated with streptozotocin for 18 h or $H_2O_2$ (10 μM) for 12 h. Cells were washed 3 times with 1xPBS and then cells were fixed with 0.3% formaldehyde. Among the cells overexpressed in GFP-Parkin, only the cells simultaneously expressed with HA-Plk3 were observed under a fluorescence microscope and quantified. The experiment was conducted three times independently. Intracellular ROS production was measured using a fluorescence microscope (Olympus LX71 microscope), and the images were analyzed using MetaMorph software (Universal Imaging, Westchester, PA). Analysis using a fluorescent microscope was prepared in the same way as in the previous study [9, 28, 32, 33].

## 2.6. Western blot analysis and antibodies

Protein lysate samples from two different pancreatic beta-cell lines were lysed by RIPA lysis buffer on ice supplemented with protease inhibitors, including 1 mM PMSF. Protein lysate samples were separated by preparing stacking gel and 10–15% separating gel during 4–8 h, and then transferred onto PVDF membranes. The membranes were blocked in TBS-T buffer in 5% fat-free milk for one hour at room temperature. Next, the PVDF membrane was incubated with primary antibodies. The membrane was washed three times for 1% TBST and subsequently incubated for one hour with related rabbit anti-mouse IgG-HRP secondary antibodies (Abcam). Rabbit polyclonal antibodies recognizing PINK1 (ab23707) or Plk3 (PA5-97143) were obtained from Abcam or Invitrogen. Rabbit polyclonal antibody recognizing Parkin (#2132), LC3B (#2775), or VDAC (V2139) was purchased from Cell Signaling or Sigma. Anti-beta-actin mouse antibodies were purchased from Sigma. Protein change confirmation analysis using immunoblotting has been described previously [9, 28, 32–34].

## 2.7. Mitochondria and cytosolic fractionation

To confirm the protein expression of Parkin translocation to the mitochondria, Beta TC-6 mouse pancreas islet cells β cells were harvested in Mitochondria/Cytosol Fractionation Kit (Abcam, ab65320). Briefly, single pancreatic cells ($1x10^7$) from which the culture medium was cleared were placed in 1 ml of 1x Cytosol Extraction Buffer Mix (containing DTT and Protease Inhibitors) at 4˚C for 10 minutes. Single pancreatic cells were homogenized by passing through a chilled tissue grinder 50 times. At this time, it was checked whether there were any shining cells (cell membranes) from passage 40. Homogenize cells were centrifuged at $600 \times g$ (3000 rpm) to discard the pellet containing the nucleus and damaged cells, and the supernatant was collected. In the supernatant transferred to a new tube, the re-supernatant was collected

for the cytoplasmic fraction via a speed centrifuge at 10,000 x g (14,000 rpm) for 30 min. Here, the pellet was solubilized with 100 μl of the Mitochondrial Extraction Buffer Mix (containing DTT and protease inhibitors) for mitochondrial fractionation. VDAC antibody was used as a marker of mitochondria and tubulin antibody was used as a cytoplasmic marker.

## 2.8. 3D organoid assay and insulin measurement

For 3D organoid assays, mouse islet Beta TC-6 or Beta TC-tet cells were cultured on NanoCulture plates (Scivax, Japan). Pancreatic cells grown in a 2D tissue culture dish were made into a single cell state using 2xTrypsin-EDTA, and after centrifugation (600 × g, 3000 rpm), 500 cells were prepared to enter each well of the culture dish. After seeding cells, cell sphere numbers were observed and recovered for 6 or 7 days. To check whether the pancreatic cells secrete insulin under 50 nM STZ or 10 nM $H_2O_2$, pancreatic cells were cultured for 5 days in five different 3D NanoCulture plates. At the same time, cells were also treated with antioxidants (including 1 μM NAC or natural COA water) for 5 days or mitophagy inhibition factors (including 5 μM 3-MA or *Parkin* shRNA), and then cell spheres formation were observed with a microscope for insulin secretion study. Cells were incubated with 3 mM glucose or 25 mM glucose in KRBH buffer for 1 h. Glucose-stimulated insulin secretion of the supernatant medium was measured by Crystal Chem ELISA kit (Downers Grove, IL) 3D organoid study ex vivo was prepared in the same way as in the previous study [9, 33, 34]. The amount of insulin secreted by pancreatic cell spheres was normalized to total cell protein.

## 2.9. Statistical analysis

At least three independent replicates were assessed for each of the in vitro or ex vivo experiments, and pooled data were presented as mean ± standard error. Results were expressed as individual data points or as the mean ± S.D. Statistical analyses were performed using Graph-Pad Prism software (version 6.0; GraphPad Software, San Diego, CA). Statistical comparisons of scatter plots and bar graphs were performed by using either the one-way Analysis of Variance (ANOVA) or 2-way ANOVA for multiple comparisons with Holm-Sidak post hoc test. A P-value of $<0.05$ was considered significant. Statistical significance was defined as $P<0.05(*)$, $P<0.01(**)$, and $P<0.001(***)$, ns: not significant. Statistical analysis has been described previously [9, 28, 32–34].

## 3. Results

### 3.1. The mitochondrial translocation of Parkin for mitophagy is activated by mitochondrial ROS in streptozotocin (STZ)-induced diabetes stress

Since several recent studies have reported that diabetes is affected by mitophagy [5, 6, 13, 17, 22, 23, 25, 31], we hypothesized that Parkin could play an important role in mitochondria in pancreatic cells in diabetes. We observed many mitophagy-related vesicles in Beta TC-6 islet cell lines treated with streptozotocin (STZ) diabetes stress through electron microscopy (Fig 1A and 1B). These results were also found in another pancreatic cell, Beta TC-tet (Fig 1B). In addition, we confirmed that STZ stress increased the mitochondrial ROS to a level similar to that of carbonyl cyanide m-chlorophenylhydrazone (CCCP), a mitophagy inducer [6, 17, 21, 27–29, 32] (Fig 1C and 1D). However, no changes in the expression of endogenous Parkin and PINK1 proteins in pancreatic β cells were observed with both STZ and CCCP treatments, although increased Cleavage of LC3 (LC-3II) expression was seen (Fig 1E). As the translocation of Parkin to depolarized mitochondria for mitophagy in cells damaged by ROS occurs in various cells [2, 4–6, 8, 10–14, 16, 19, 25, 26, 31], we monitored the protein movement of

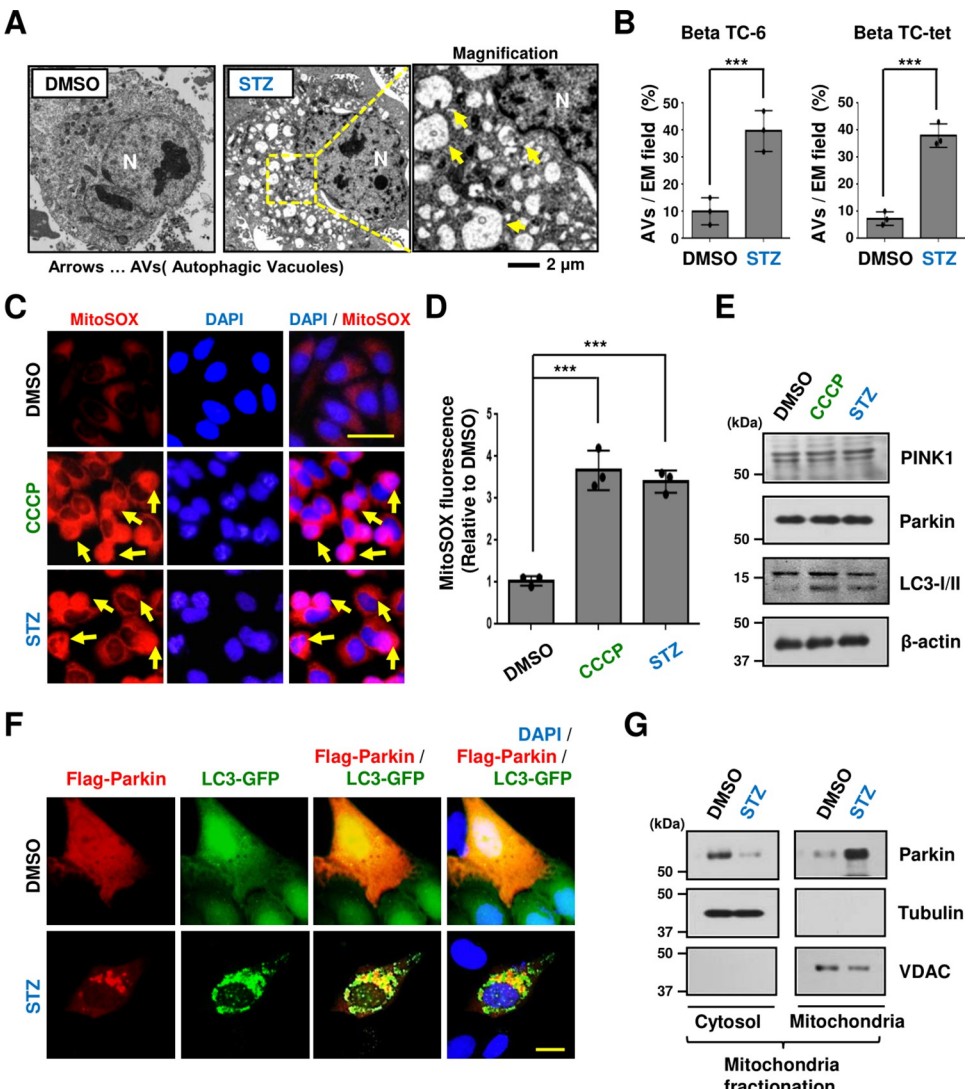

**Fig 1. The mitochondrial translocation of Parkin for mitophagy is activated by mitochondrial Reactive Oxygen Species (ROS) in streptozotocin (STZ)-induced diabetes stress.** (A, B) Beta TC-6 or Beta TC-tet cells were treated with streptozotocin (STZ, 50 μM) for 72 h. Cells were fixed with glutaraldehyde and then cut cell sections were observed with an electron microscope (A) and quantified (B). Scale bars, 2 μm. (C, D, E) Beta TC-6 cells were treated with STZ (50 μM) for 12 h or Carbonyl cyanide 3-chlorophenylhydrazone (CCCP, 10 μM) for 4 h. Cells were stained with MitoSOX (5 μM) for 30 min, and cells were observed under a fluorescence microscope (C) and quantified (D). Cell lysates were analyzed by immunoblotting with the antibodies indicated (E). Quantification of PlNK1, and Parkin levels relative to β-actin levels. (F) Cells were transfected with the indicated plasmids and then cells were treated with streptozotocin and observed by fluorescence microscopy. (G) Beta TC-6 cells were treated with 50 μM STZ for 72 h. Cells homogenized with a grinder were classified into mitochondria and cytoplasm, and then protein changes were confirmed by immunoblotting with the antibodies indicated. ***$P < 0.001$. All experiments (IF) were repeated independently at least three times with similar results. Representative images, scale bar, 20 μm. All statistical comparison of scatter plot and bar graph was performed by repeated measure ANOVA with multiple comparisons test.

Parkin under STZ diabetic stress. Parkin overexpressed in Beta TC-6 cells in general culture condition was blurred throughout the cells but aggregated in the form of Parkin-specific dots (Parkin puncta formation) [21, 27] in STZ-treated cells. (Fig 1F). Since CCCP induces a marked translocation of Parkin from the cytoplasm to the mitochondria [6, 17, 21, 27], it appears that STZ also translocates Parkin to depolarized mitochondria because punctuate

staining of autophagosome-localized LC3-II in pancreatic cells under STZ treatment was observed at the same location as Parkin protein. In the mitochondrial fractionation assay, we also verified that Parkin protein expression was more detected in mitochondrial samples under STZ stimulation compared to controls, indicating that Parkin was translocated from the cytoplasm to mitochondria under diabetic stress (Fig 1G). These results suggest that Parkin is recruited to mitochondria using mitochondrial ROS for mitophagy under STZ diabetes stress.

## 3.2. PINK1 regulates mitochondrial recruitment of Parkin through mitochondrial ROS in diabetes stress

Previous studies have reported that activated PINK1, by mitochondrial damage, accumulates in the outer mitochondrial membrane and phosphorylates Parkin, thereby helping the recruitment of Parkin to dysfunctional mitochondria [5, 21, 22, 25–28, 32]. This induces intracellular mitophagy by the E3 ubiquitin ligase activity of Parkin [21, 22, 25–28, 32]. First, we checked whether the STZ stress-induced mitochondrial ROS induction is regulated by PINK1. Deletion of *PINK1* (Fig 2A, S1 Fig in S1 File) had no effect on mitochondrial ROS in Beta TC-6 pancreatic cells compared to Control (Fig 2B and 2C). STZ-treated *PINK1*-deleted cells were also similar to STZ-treated alone. Furthermore, we confirmed that cells treated with Hydrogen Peroxide ($H_2O_2$), a key contributor to intracellular ROS-dependent signaling, also showed results like that of STZ stimulation conditions (Fig 2C). However, interestingly, the deleted *PINK1* gene suppressed the translocation of Parkin to the mitochondria via STZ stress-induced mitochondrial ROS (Fig 2D and 2E). The same effect of deletion of *PINK1* was also shown in other pancreatic cells, Beta TC-tet (Fig 2E). Using mitochondrial fractionation assay, we further confirmed whether the translocation of Parkin to mitochondria is regulated by PINK1. As shown in Fig 2F, Parkin protein was highly expressed in STZ-treated control shRNA (as Fig 1G), whereas it was significantly reduced in PINK1-deleted cells (Fig 2F). Taken together, these data suggest that PINK1 induces mitochondrial translocation of Parkin by diabetic stress-related mitochondrial ROS.

## 3.3. FOXO3A block mitochondrial recruitment of Parkin by negatively regulating Plk3 increased by diabetic stress

To inhibit the mitochondrial recruitment of Parkin in abnormal pancreatic cells, we investigated how to reduce the mitochondrial ROS amplified by diabetic stress. In S2A Fig in S1 File, we found that N-acetyl-l-cysteine (NAC), a ROS scavenger [7, 8, 10, 21, 26], significantly inhibited mitochondrial ROS in STZ-treated Beta TC-6 cells. Also, we have previously reported that natural COA water suppresses the generation of unstable mitochondrial ROS by negatively blocking STZ-induced Plk3 expression in pancreatic β cells [9], confirming that this effect is very comparable to that of NAC (S2A Fig in S1 File). We found that this antioxidant effect was equally suppressed in PINK1 gene overexpressed cells, suggesting that PINK1 is directly affected by mitochondrial ROS regulation. Furthermore, we hypothesized that antioxidants would have a negative effect on the mitochondrial translocation of Parkin stimulated by STZ diabetic stress-mediated mitochondrial ROS. The recruitment of Parkin in mitochondria in STZ-treated cells was dramatically prevented by both NAC and natural COA water (Fig 3A), suggesting that Parkin translocation is mitochondrial ROS-dependent. Interestingly, PINK1 overexpressing cells were not at all affected by antioxidant treatment, and Parkin's mitochondrial recruitment proceeded smoothly. These results suggest that PINK1 is activated regardless of the presence of upstream mitochondrial ROS, as it is already activated intracellularly by PINK1 gene overexpression. This means that this diabetic stress process leads to the mitochondrial ROS-PINK1-Parkin axis. Here, we could understand that the mitochondrial

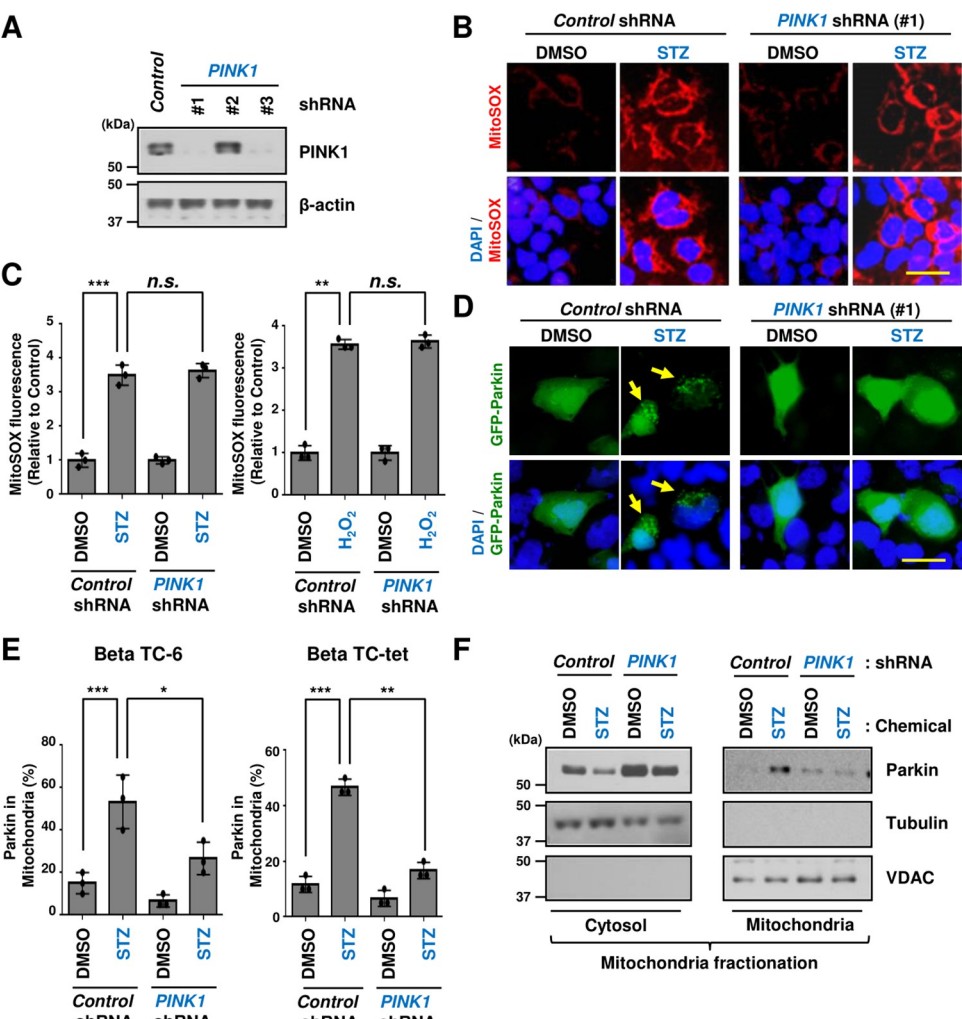

**Fig 2. PINK1 regulates mitochondrial recruitment of Parkin through mitochondrial ROS in diabetes stress.** (A) Beta TC-6 cells were infected with the indicated plasmids and then cells were observed for changes in PINK1 protein by immunoblotting. (B, C, D, E) Beta TC-6 cells or Beta TC-tet were infected with the indicated and then cells were treated with streptozotocin for 18 h or $H_2O_2$ (10 μM) for 12 h. Cells were stained with MitoSOX (5 μM) for 30 min and then cells were analyzed by immunofluorescence assay (B) and quantified (C). Beta TC-6 cells or Beta TC-tet were transfected with GFP-Parkin and then cells were infected with the indicated plasmids (D, E). After 24h, Cells were treated with streptozotocin for 18 h, and then cells were observed by fluorescence microscopy (D) and quantified (E). Representative images, scale bar, 20 μm. Statistical comparison of scatter plot and bar graph was performed by repeated measure ANOVA with multiple comparisons test; *$P < 0.05$, **$P < 0.01$, ***$P < 0.001$, *n.s.*, non-specific. All experiments were repeated independently at least three times with similar results. (F) Beta TC-6 cells were infected with the indicated plasmids and then cells were treated with streptozotocin for 24 h, and then cells were then observed for protein changes by immunoblotting after mitochondrial fractionation.

ROS signaling system must precede the migration of PINK1 and Parkin to the outer mitochondrial membrane. As a previous study reported that STZ stress increases mitochondrial ROS through intracellular Plk3 gene activity [9], we investigated whether Plk3 is also involved in the mitochondrial recruitment of Parkin. As shown in S2B Fig in S1 File, we first confirmed that overexpression of Plk3 increased mitochondrial ROS in beta cells and this also showed the effect of recruiting Parkin to the mitochondria (Fig 3B). However, antioxidants including NAC and natural COA water completely blocked the ROS effects of Plk3 (S2B Fig in S1 File). Plk3-overexpressing pancreatic beta cells induced the mitochondrial ROS or Parkin's

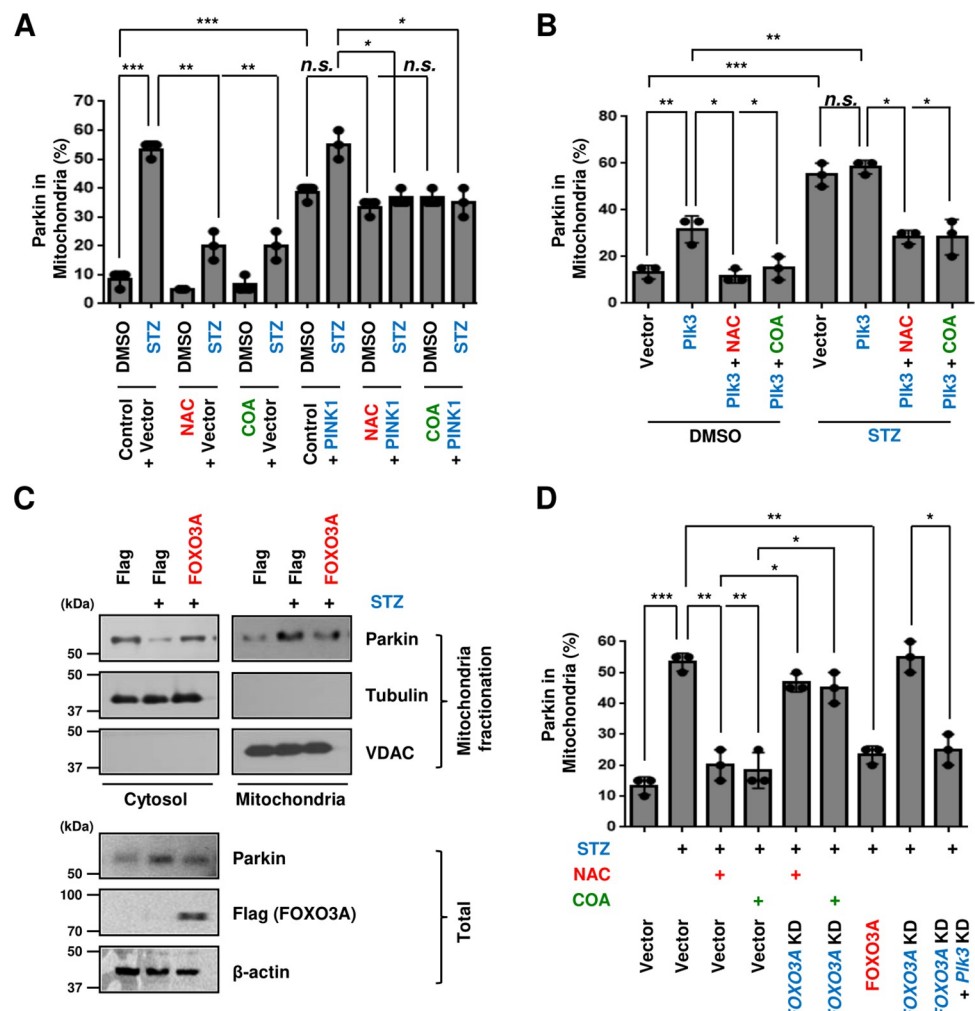

**Fig 3. FOXO3A block mitochondrial recruitment of Parkin by negatively regulating Plk3 increased by diabetic stress.** (A, B, D) GFP-Parkin overexpressed Beta TC-6 cells were treated with streptozotocin for 12 h and then cells were transfected with the indicated plasmids or treated with antioxidants including NAC (10 mM) or natural COA water for 12 h. Cells were analyzed by immunofluorescence assay and quantified. Statistical comparison of scatter plot and bar graph was performed by repeated measure ANOVA with multiple comparisons test; *P < 0.05, **P < 0.01, ***P < 0.001, n.s., non-specific. All experiments were repeated independently at least three times with similar results. (C) Beta TC-6 cells were transfected with the indicated plasmids and then cells were treated with streptozotocin for 24 h, and then cells were then observed for protein changes by immunoblotting after mitochondrial fractionation.

mitochondrial recruitment in higher amounts when treated with STZ than Plk3 alone, however, these increased amounts were the same as when treated with STZ alone (S2B Fig in S1 File and Fig 3B) indicating that it is an effect of STZ rather than an effect increased by the overexpression of the Plk3 gene. Meanwhile, both antioxidants effectively suppressed all STZ stress-Plk3 effects, such as mitochondrial ROS generation and mitochondrial translocation of Parkin (S2B Fig in S1 File and Fig 3B). Collectively, these results demonstrate that Plk3 is the main regulator of the recruitment of Parkin to the mitochondrial membrane, including PINK1, which induces mitochondrial ROS under STZ diabetic stress. Taken together, it can be suggested that the diabetic stress signal has the Plk3-mtROS-PINK1-Parkin pathway. A recent study reported that transcription factor FOXO3A is closely related to autophagy proteins in somatic cells of various organs, including stem cells, and that it regulates ROS-related

metabolism by regulating mitochondrial genes [35–37]. We first confirmed that overexpression of the FOXO3A gene significantly inhibited the increase of mitochondrial ROS induced by STZ stress (S3 Fig in S1 File). We also found that this role of FOXO3A also had the effect of significantly reducing Parkin mitochondrial recruitment when compared to STZ-treated cells in mitochondrial fractionation and immunofluorescence assays (Fig 3C and 3D). Meanwhile, the deficiency of *FOXO3A* does not appear to decrease the mitochondrial translocation of Parkin even after the treatment with antioxidants (Fig 3D), suggesting that the presence of FOXO3A is important for antioxidants for their role. In addition, co-deletion of *FOXO3A* and *Plk3* genes again reduced mitochondrial recruitment of Parkin more significantly compared to the *FOXO3A* gene alone (Fig 3D), suggesting that FOXO3A blocks mitochondrial recruitment of Parkin by negatively regulating Plk3 increased by diabetic stress.

### 3.4. The decrease in insulin secretion of pancreatic beta cells derived from mitophagy of Parkin in the 3D organoid (ex vivo) model is compensated by antioxidants or FOXO3A

To investigate whether pancreatic β cells secrete insulin in an *ex vivo* model, we cultured each different 3D organoid nanomaterial dishes [38, 39] from groups treated with various diabetes-induced stresses. First, we confirmed the growth of pancreatic β islet cells in the 3D *ex vivo* culture, and these cells began to take on a spherical shape after 5 days of culture (Fig 4A). As shown in Fig 4B and 4C, STZ damage blocked the growth of pancreatic β cells in the 3D organoid model, and this cell growth inhibition was also the same for $H_2O_2$. However, two antioxidants, NAC and natural COA water significantly protected the growth of STZ- or $H_2O_2$-damaged cells, suggesting that stabilized ROS in cells positively affects beta cell proliferation and maintenance (Fig 4B and 4C). These effects were also found in *Parkin* shRNA and 3-Methyladenine (3-MA), a mitophagy inhibition factors, and overexpression of FOXO3A suggesting that mitophagy negatively affects the growth of pancreatic cells, similar to the function of antioxidants In a previous study, we already reported that natural COA water significantly restored the restricted insulin secretion of pancreatic cells by STZ diabetic stress stimulation [9]. In 3D *ex vivo* culture (Fig 4D), most of the pancreatic cells stimulated with a low concentration of glucose (3 mM glucose) secreted very small amounts of insulin, but a large amount of insulin was measured in the group treated with a high concentration of glucose (25 mM glucose). We found that insulin secretion by high-concentration glucose stimulation in the supernatant medium of STZ-treated pancreatic β cells was significantly lower than that of control pancreatic cells and that this decrease in insulin secretion was restored by two antioxidants, NAC or natural COA water (Fig 4D). Similarly, the recovery of insulin secretion in beta cells decreased by STZ treatment was similarly confirmed in the loss of the *Parkin* gene, overexpression of FOXO3A, 3-MA treatment. Collectively, these data indicate that restricted mitochondrial ROS and mitophagy processes are required for stable cell proliferation and insulin secretion of pancreatic cells. From the above, we suggest that blocking the Plk3-Parkin-mediated mitophagy using antioxidants or FOXO3A gene might be one of the future diabetes treatment and prevention methods (Fig 4E).

## 4. Discussion

Parkin is a vital regulator and core organizer of mitophagy for the control of intracellular mitochondrial quality [17, 21, 22, 25–29, 32]. As the role of Parkin's mitophagy in recently reported diabetic cardiomyopathy (DCM) or diabetes-associated submandibular gland (SMG) disease are very controversial [5, 17, 22, 23, 31], future studies on diabetes related to Parkin are needed and this study is important in that regard. Although mitophagy-limited studies of the major

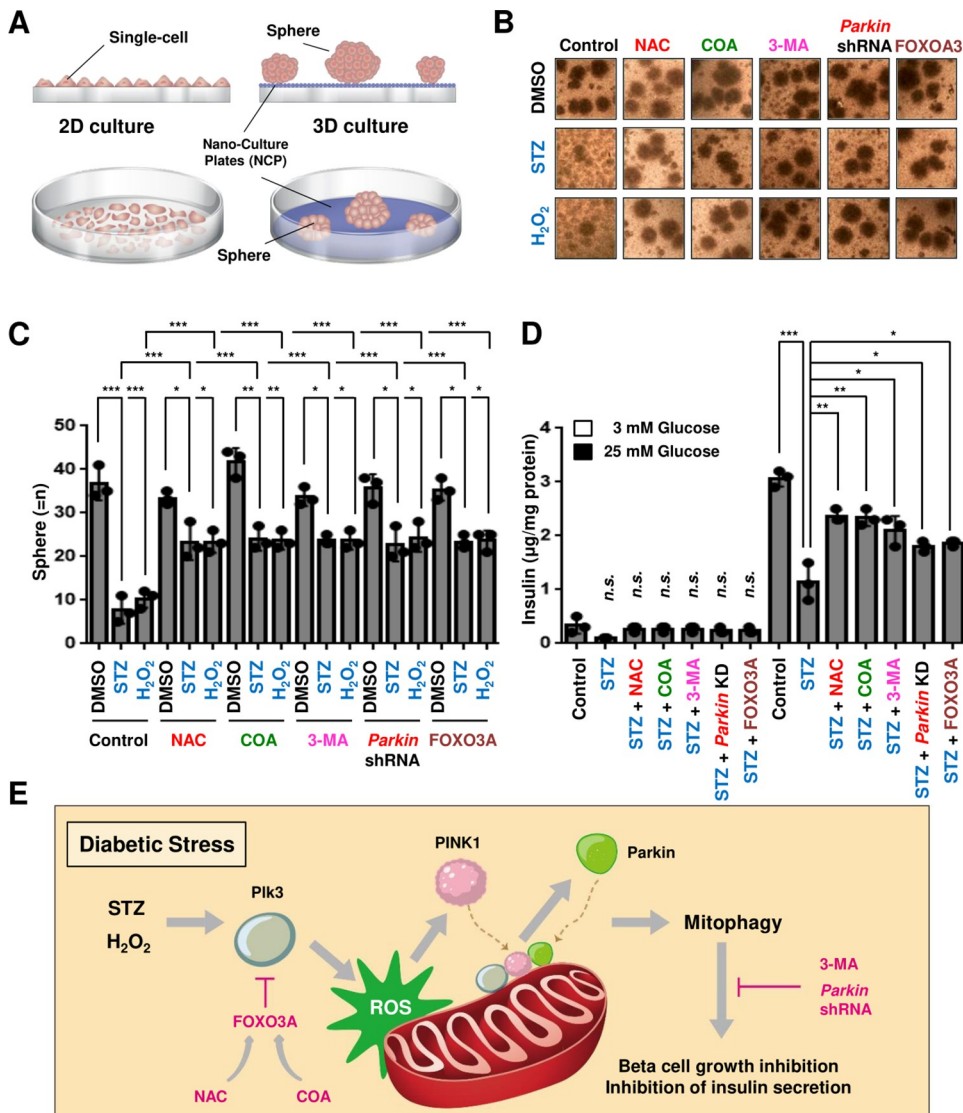

**Fig 4. The decrease in insulin secretion of pancreatic beta cells derived from mitophagy of Parkin in the 3D organoid (*ex vivo*) model is compensated by antioxidants or FOXO3A.** (A) The schematic model for experimental cell culture design. Unlike 2D (left), 3D organoid cell culture (right) contains special nanomaterial at the bottom of the culture dish, so the cells grow more stably and are almost formed in the shape of real cells like human organs or cells. (B, C) After 2 days of incubation in a 3D Nano-Culture Plate (NCP) dish, Beta TC-6 cells were treated with streptozotocin (5 µM) or $H_2O_2$ (1 µM) for 5 days. At the same time, cells were also treated with antioxidants (including 1 µM NAC or natural COA water), mitophagy inhibition factors including 5 µM 3-MA, *Parkin* shRNA, or FOXO3A for 5 days, and then cell spheres were observed with a microscope (B) and quantified (C). (D) Cells were incubated with 3 mM glucose or 25 mM glucose in KRBH buffer for 1 h. Glucose-stimulated insulin secretion of the supernatant medium in each classified cultured islet sphere sample (*n = 3* per group) was measured by ELISA kit. Insulin secretion was normalized to total protein content. All statistical comparison of scatter plot and bar graph was performed by repeated measure ANOVA with multiple comparisons test; *$P < 0.05$, **$P < 0.01$, ***$P < 0.001$. All experiments were repeated independently at least three times with similar results. (E) The schematic model. Parkin is a key regulator of mitophagy-mediated pancreatic cell damage by mitochondrial translocation. The mitophagy role of the PINK1-Parkin axis is regulated by the increase in mitochondrial ROS induced by Polo-like kinase 3 (Plk3). Transcription factor FOXO3A induced by antioxidants directly inhibits Plk3, thus suppressing the ultimate role of mitophagy in STZ-treated pancreatic cells. FOXO3A as a negative feedback loop blocks mitochondrial recruitment of Parkin by negatively regulating Plk3 increased by diabetic stress, thereby stabilizing pancreatic cells.

role of Parkin have been successful [13, 40], we have shown that Parkin also functions as a novel E3 ligase for regulating the Cdc20/Cdh1 complex for cell division [32] and inhibits the function in colon inflammation-mediated necroptosis [28]. More specific studies of Parkin in diabetes are required. Our data showed that the expression of Parkin protein was not high in pancreatic cells compared to other organ normal cells and there was no protein change even under STZ treatment. However, Parkin was recruited to the mitochondria by diabetic stress stimulation, which played a key role in mitophagy function (Figs 1 and 2). We also found for the first time that Parkin-mediated mitophagy induced by diabetic stress inhibits pancreatic cell growth and insulin secretion in 3D organoid *ex vivo* model (Fig 4). Collectively, STZ diabetes-induced stimulation causes mitochondrial translocation of Parkin which is initiated by the Plk3-mtROS-PINK1 signal pathway (Figs 2 and 3). Notably, a series of processes mediated by Parkin in the long term is perceived as negative stimuli, at least in pancreatic cells, and this signaling mechanism selects processes that damage the cell itself rather than protect itself. One could suggest that mitophagy in the early stages of diabetes in pancreatic cells may be protective for cell growth but long-term mitophagy (i.e. chronic) diabetes-induced stress seems to inhibit cell growth. Also, mitochondrial ROS are closely related to metabolic diseases such as diabetes and diabetes-related complications [1, 4, 7, 8, 11, 12, 14, 19, 20, 26]. In this case, Parkin's mitochondrial recruitment is very important, and the increase in mitochondrial ROS caused by Plk3 might play a key role in PINK1 activation (Fig 3D). And antioxidants, such as NAC and natural COA water, control mitochondrial ROS by diabetes-induced damage and negatively regulate the Parkin-mediated mitophagy function by PINK1 in the mitochondrial outer membrane. Through this study, we suggested the possibility that high mitophagy activity by mitochondria ROS may play a key role in the development of diabetes through the inhibition of beta cell growth and decreased insulin secretion (Fig 4).

The association between Plk3 and diabetes has already been reported in diabetes-related cataracts [9, 41], but its mechanism and other studies have not been specifically reported. We found that Plk3 as a novel key regulator of diabetes mellitus by increasing mitochondrial ROS [9] and newly discovered that it is an early initiator regulating mitophagy (Fig 3). In this regard, the identification of the role of Plk3 in diabetes is expected to be a new starting point in future diabetes research. We also found that the FOXO3A gene negatively regulates Plk3 through this study. Although several studies have reported that FOXO3A plays an important role in the regulation of mitochondrial ROS metabolism and autophagy [35–37], there has been no report on the relationship between FOXO3A and Plk3 or FOXO3A and Parkin's mitochondrial recruitment. We first found that FOXO3A directly prevents mitophagy by blocking Plk3 when pancreatic cells are subjected to STZ diabetic stress in several experiments (Fig 3). This event blocks the mitochondrial recruitment of Parkin, which is regulated by Plk3. Furthermore, *FOXO3A* is a transcription factor so it would most likely repress Plk3 at the transcriptional level rather than the translational level even though further study would be needed to confirm this. In summary, FOXO3A causes a negative feedback loop by inhibiting the Plk3-PINK1-Parkin axis for mitophagy when diabetic damage is stimulated in pancreatic cells.

To obtain more accurate and practical research results, we introduced the *ex vivo* technique in this study. 3D *ex vivo* can implement cellular tissue most like human tissue, making it possible to more accurately predict the effects of various drugs and compounds [9, 33, 34, 38, 39]. In this experiment, we focused more on how mitochondrial ROS and mitophagy increased by STZ-induced diabetic damage affects pancreatic cell growth and insulin secretion (Fig 4). We could observe more clearly in the 3D experiment that the growth of pancreatic cells was inhibited due to the excessive secretion of mitochondrial ROS by STZ. Furthermore, it was also confirmed that the generation of mitophagy was totally blocked by inhibiting these mitochondrial ROS. In particular, the inhibition of the *Parkin* gene can be a new alternative as an effective

diabetes suppression that can significantly restore the decrease in insulin secretion caused by diabetic stress (Fig 4). As such, the 3D organoid cell culture technology has been verified as an innovative experimental verification tool, so further in-depth research is needed. In particular, type 2 diabetes (T2D) is a disease in which morbidity can be prevented if it can be diagnosed and treated at an early stage, therefore, it is necessary to develop a biomarker that can be applied to diabetic patients customized to identify the gene. FOXO3A, Plk3, Parkin, and PINK1 genes may be used in future precision medicine as biomarkers for diabetes. The genetic precision mapping found in this way may be able to provide patients with higher accuracy in the diagnosis process of early diabetes and complications prediction in future precision medicine. Of course, several challenges remain, such as clinical verification of whether these genes have high gene expression in different diabetic patient samples. In addition, there are still many studies to be done, such as the functional studies in pancreatic cells as E3 ligase of Parkin, the Plk3 regulation mechanism of ROS inhibitors (NAC, Trolox or natural COA water, etc.) or FOXO3A, and the insulin secretion process inhibition in the mitophagy process.

## Supporting information

**S1 File. Contains detailed information about the S1-S3 Figs.**
(PDF)

**S1 Data.**
(XLSX)

**S1 Raw images.**
(PDF)

## Acknowledgments

We thank Department of Radiology (Mayo Clinic, Rochester, MN), College of Nursing (Dankook University, Korea), Department of Internal Medicine (Chonnam National University Medical School, Korea), and U&Hang Clinic (Korea) for supporting experimentation and data analysis. We thank Christine Song (Mayo High School, Rochester, MN) for editing this paper.

## Author Contributions

**Conceptualization:** Ji Yeon Shim, Jin Ook Chung, Val J. Lowe, SeungBaek Lee.

**Data curation:** Ji Yeon Shim, Jin Ook Chung, Dawa Jung, Pil Soo Kang, Ayse Tuba Kendi, SeungBaek Lee.

**Formal analysis:** Jin Ook Chung, Dawa Jung, Pil Soo Kang, Ayse Tuba Kendi, SeungBaek Lee.

**Funding acquisition:** Val J. Lowe, SeungBaek Lee.

**Investigation:** Ji Yeon Shim, Pil Soo Kang, Seon-Young Park.

**Methodology:** Ji Yeon Shim, Jin Ook Chung, Dawa Jung, Seon-Young Park, SeungBaek Lee.

**Project administration:** Val J. Lowe, SeungBaek Lee.

**Resources:** Val J. Lowe.

**Supervision:** Val J. Lowe, SeungBaek Lee.

**Validation:** Ji Yeon Shim, Jin Ook Chung, Dawa Jung, Seon-Young Park, Ayse Tuba Kendi, Val J. Lowe, SeungBaek Lee.

**Visualization:** Ji Yeon Shim, Dawa Jung, Ayse Tuba Kendi, SeungBaek Lee.

**Writing – original draft:** Ji Yeon Shim, Jin Ook Chung, Dawa Jung.

**Writing – review & editing:** Ayse Tuba Kendi, Val J. Lowe, SeungBaek Lee.

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
