## [Decision Letter · Decision Letter 0]

7 Dec 2022

PONE-D-22-27165Parkin-mediated mitophagy is negatively regulated by FOXO3A, which inhibits Plk3-mediated mitochondrial ROS generation in STZ diabetic stress-treated pancreatic β cellsPLOS ONE

Dear Dr. Lee,

Thank you for submitting your manuscript to PLOS ONE. After careful consideration, we feel that it has merit but does not fully meet PLOS ONE’s publication criteria as it currently stands. Therefore, we invite you to submit a revised version of the manuscript that addresses the points raised during the review process. Please submit your revised manuscript by Jan 21 2023 11:59PM. If you will need more time than this to complete your revisions, please reply to this message or contact the journal office at plosone@plos.org. Please include the following items when submitting your revised manuscript:

We look forward to receiving your revised manuscript.

Kind regards,

Prasanth Puthanveetil

Academic Editor

PLOS ONE

Journal Requirements:

2. Please note that PLOS ONE has specific guidelines on code sharing for submissions in which author-generated code underpins the findings in the manuscript. In these cases, all author-generated code must be made available without restrictions upon publication of the work. Please review our guidelines at https://journals.plos.org/plosone/s/materials-and-software-sharing#loc-sharing-code and ensure that your code is shared in a way that follows best practice and facilitates reproducibility and reuse. New software must comply with the Open Source Definition.

“V.L., and S.B.L. supervised the project. This work was also supported by grant number E&P-RES20191001-02 to S.B.L. and by the Dorothea Berggren Charitable Foundation to V.L.”

“Dr. Lowe is a consultant for AVID Radiopharmaceuticals, Eisai Co. Inc., Bayer Schering Pharma, GE Healthcare, and Merck Research, and receives research support from GE Healthcare, Siemens Molecular Imaging, AVID Radiopharmaceuticals, and NIH (NIA, NCI). Dr. Lee reported grants from E&P Co., Ltd. (South Korea) during the conduct of the study.”

7. PLOS ONE now requires that authors provide the original uncropped and unadjusted images underlying all blot or gel results reported in a submission’s figures or Supporting Information files. This policy and the journal’s other requirements for blot/gel reporting and figure preparation are described in detail at https://journals.plos.org/plosone/s/figures#loc-blot-and-gel-reporting-requirements and https://journals.plos.org/plosone/s/figures#loc-preparing-figures-from-image-files. When you submit your revised manuscript, please ensure that your figures adhere fully to these guidelines and provide the original underlying images for all blot or gel data reported in your submission. See the following link for instructions on providing the original image data: https://journals.plos.org/plosone/s/figures#loc-original-images-for-blots-and-gels.

Additional Editor Comments:

Dear Authors,

We are writing to inform you that both the reviewers have demonstrated enthusiasm in your work. There are certain concerns that need to be addressed before it can be in the publishable form.

Please address all the concerns raised by our reviewers in your revised submission.

We will be waiting for the revised manuscript.

Best Wishes,

Academic Editor

Reviewers' comments:

Reviewer's Responses to Questions

**Comments to the Author**

1. Is the manuscript technically sound, and do the data support the conclusions?

Reviewer #1: Yes

Reviewer #2: Yes

2. Has the statistical analysis been performed appropriately and rigorously? 

Reviewer #1: Yes

Reviewer #2: Yes

3. Have the authors made all data underlying the findings in their manuscript fully available?

Reviewer #1: Yes

Reviewer #2: No

4. Is the manuscript presented in an intelligible fashion and written in standard English?

Reviewer #1: No

Reviewer #2: No

5. Review Comments to the Author

Reviewer #1: This study titled ‘Parkin-mediated mitophagy is negatively regulated by FOXO3A, which inhibits Plk3-mediated mitochondrial ROS generation in STZ diabetic stress-treated pancreatic β cells’ was aimed at investigating FOXO3A activities in the negative regulation of Parkin-mediated mitophagy by inhibiting Plk3.

The introduction was concise; it provides background and includes relevant references, the first reference in the introduction has ‘[references]. [1-18]’, this is a typo. The research design is appropriate, and the methods adequately described. The results were clearly presented, and the conclusions supported by the results.

The abstract needs to be reworded. Please ensure all abbreviations are appropriately defined at first use.

The authors use a couple of third person pronoun ‘We’, not sure if that is ok. This could be reworded. The significance of this study needs to be stress more in conclusion. Grammar, spelling, and typos need to be checked.

Reviewer #2: In this manuscript, Shim et al describe regulation of Parkin translocation to mitochondria in the context of STZ treatment. Through genetic and pharmacological approaches, the authors connect STZ effects to Parkin mitochondrial translocation via mitochondrial ROS and several intermediate mechanisms involving FOXO3A, Plk3, and PINK1. These experiments are generally well-designed, and the data are robust, but some results are over-interpreted. One limitation that is not discussed is that the authors conflate mitophagy with mitochondrial translocation of Parkin. The authors should either bolster these experiments with complementary readouts of mitophagy or soften their language and point out this limitation. Finally, in many areas there is ambiguous or confusing language in the text, and the revised manuscript should be carefully edited. Nevertheless, the manuscript would be acceptable for publication if these issues were rectified.

Major comments:

1. Figure 3A. The authors claim “PINK1 overexpressing cells were not at all affected by antioxidant treatment”. There is no control showing PINK1-overexpressing cells (with and without STZ) in the absence of antioxidants that would be necessary to make this claim. It remains possible STZ increases Parkin translocation to mitochondria in PINK1-overexpressing cells in the absence of antioxidant treatment (and that this could be affected by antioxidants).

2. Figure 4D. Are differences in insulin secretion simply a result of differences in cellular insulin content? As insulin released into the medium appears to mirror sphere numbers (From Fig. 4C), the authors should also show insulin secretion normalized to insulin content to distinguish secretory defects from differences in cell number or insulin production.

Minor comments:

1. Figure 4C. It is claimed that “NAC and natural COA water significantly protected the growth of STZ- or H2O2-damaged cells, suggesting that stabilized ROS in cells positively affects beta cell proliferation and maintenance”, however the authors do not measure survival/death or proliferation (and sphere numbers are only shown for one time point) so the extent that each feature is affected and contributes to an increase in sphere number is unclear. The authors should soften the language or add relevant measurements.

2. The two panels shown in Figure 2C need to be labeled. Are these panels different cell lines?

3. “In Figure 3A, we found that NAC, a ROS scavenger, significantly inhibited mitochondrial ROS in STZ-treated Beta TC-6 cells”. This claim appears to refer to the experiment in Supplementary Figure S2A.

4. The text needs to be edited for clarify throughout, including but not limited to the below statements:

-Page 10: “…suggesting that PINK1 is a mitochondrial ROS-regulated downstream”. This appears to be a sentence fragment.

-Page 11: “the effect of Plk3 is within STZ stimulation”. What do the authors mean by “within STZ stimulation?”

-Page 12: “natural COA water under STZ diabetic stress stimulation positively prevents the inhibition of insulin secretion”. The term “positively prevents” is confusing.

-Page 12: “…treated with a large amount of glucose”. The appropriate term is “a high concentration” rather than “a large amount” of glucose.

-Page 12: “cells cultured in two antioxidants, NAC or natural COA water, still stably produced insulin in a similar amount to the control”. The authors should distinguish insulin secretion from insulin production, and it is unclear whether cellular insulin content was measured in Figure 4D to do so (see major comment 2 above).

-Page 13: “Notably, the role of Parkin’s mitophagy in our study”. Please clarify “Parkin’s mitophagy” or rephrase.

-Page 13: “FOXO3A directly blocks mitophagy by negatively blocking Plk3”. This statement would be clarified by removing the term “negatively”, or rephrasing.

5. The authors' response to the question “Describe where the data may be found” in the manuscript information table is as follows: “The data worked in the Mayo Clinic cannot be accessed and shared anywhere”. This appears to be an extraordinary exception, particularly when no data impacting patient privacy is used in this study. Such an institutional policy seems unlikely, as other investigators at this institution have deposited datasets into relevant databases (e.g., PMID: 34910509).

6. PLOS authors have the option to publish the peer review history of their article (what does this mean?). If published, this will include your full peer review and any attached files.

Reviewer #1: No

Reviewer #2: No

---

## [Author Response · Author response to Decision Letter 0]

19 Jan 2023

In the document entitled 'PONE-D-22-27165-R2', we have responded to all the reviewers' comments in detail. Thank you again for the minor revisions.

---

## [Editor Report · Decision Letter 1]

25 Jan 2023

Parkin-mediated mitophagy is negatively regulated by FOXO3A, which inhibits Plk3-mediated mitochondrial ROS generation in STZ diabetic stress-treated pancreatic β cells

PONE-D-22-27165R1

Dear Dr. Lee,

We’re pleased to inform you that your manuscript has been judged scientifically suitable for publication and will be formally accepted for publication once it meets all outstanding technical requirements.

Kind regards,

Prasanth Puthanveetil

Academic Editor

PLOS ONE

Additional Editor Comments (optional):

Dear Authors,

Thank you for responding to all the reviewers concerns.

We are delighted to accepted your work.

Please wait for the editorial and publishing staff's response for the next steps.

Looking forward to your future submissions.

Thanks,

Dr. Puthanveetil
---

## [Editor Report · Acceptance letter]

24 Apr 2023

PONE-D-22-27165R1 

Parkin-mediated mitophagy is negatively regulated by FOXO3A, which inhibits Plk3-mediated mitochondrial ROS generation in STZ diabetic stress-treated pancreatic β cells 

Dear Dr. Lee:

I'm pleased to inform you that your manuscript has been deemed suitable for publication in PLOS ONE. Congratulations! Your manuscript is now with our production department. 

Kind regards, 

on behalf of

Dr. Prasanth Puthanveetil 

Academic Editor

PLOS ONE